# Gradients protection in federated learning for Biometric authentication model

## Abstract

In federated learning (FL) environments, biometric authentication systems encounter a distinct challenge: safeguarding user privacy without sacrificing the precision necessary for identity confirmation. Although previous FL privacy research has primarily addressed broad-spectrum protections, this paper concentrates on the particular weaknesses of biometric authentication models, especially those susceptible to gradient inversion and deep gradient leakage (DGL) attacks. We introduce an innovative privacy-preserving framework specifically designed for federated biometric authentication. Our approach employs a dual strategy: (1) an authentication model that is trained on both original and modified biometric samples to maintain resilience against input perturbations, and (2) a client-side obfuscation technique that alters biometric data prior to gradient computation, efficiently preventing reconstruction attempts. The obfuscation is adaptive and privacy-aware, selectively preserving critical biometric features necessary for authentication while discarding nonessential components to reduce input size and improve accuracy. Simultaneously, this process increases the gradient distance between the original and shared data, enhancing protection against reconstruction. Additionally, block-wise shuffling is employed to disrupt the semantic structure, ensuring that any reconstructed image lacks meaningful visual content. To validate its practical use, our framework is tested in a multibiometric context using facial and fingerprint information. The blockwise transformation strategy ensures superior authentication efficiency while reducing privacy risks. Experiments conducted in various adversarial FL settings reveal that our method significantly enhances defenses against reconstruction attacks, outperforming traditional measures.

## 1 Introduction

Federated Learning (FL) Rodríguez-Barroso et al. (2023) has emerged as an essential framework for decentralized machine learning, enabling collaborative model training across multiple devices while keeping raw data local. This paradigm is particularly important in domains involving sensitive biometric information, such as facial recognition and fingerprint matching. However, recent studies have revealed significant privacy risks in FL systems. Specifically, shared gradients, even without raw data, can leak sensitive information, enabling adversaries to reconstruct original inputs and reveal identifiable biometric traits Geiping et al. (2020); Zhu et al. (2019); Zhao et al. (2020); Melis et al. (2019); Shokri et al. (2017).

Advances in gradient inversion attacks have exacerbated these concerns. For example, Wu et al. (2023) demonstrated that simple adaptive attacks can bypass current defenses, while Dimitrov et al. (2024) introduced SPEAR, a technique capable of reconstructing batches of images from gradients, even at large batch sizes. These results highlight the need for more robust privacy mechanisms in FL, especially for biometric authentication where identity leakage poses serious risks.

In response, we propose a privacy-preserving FL architecture designed specifically for biometric authentication. Our system consists of two modular components: (1) a client-side perturbation strategy that obfuscates input images using saliency-aware block transformations, and (2) an authentication model that verifies identity based on the perturbed data. By keeping unmodified biometric data on the client, our architecture minimizes the exposure of raw features during model training.

Unlike traditional pixel-wise perturbation, our approach operates at the block level, guided by saliency maps derived from empirical analysis. This enables efficient removal of non-essential features while preserving identity-critical information, ensuring high utility even in constrained environments such as mobile devices.

Although many real-world systems still rely on unimodal biometric authentication, dual-modality systems, combining modalities like face and fingerprint, are gaining adoption in high-security applications such as border control, forensics, and financial services Bala et al. (2022); Kazi et al. (2024). These systems offer greater robustness and resistance to spoofing attacks, and our proposed framework is explicitly designed to support such multi-modal biometric fusion. Our evaluation includes experiments on both individual and combined modalities, aligning with trends in real-world deployments.

To validate the effectiveness of our method, we apply it to three widely used model architectures, ResNet He et al. (2015), Vision Transformer (ViT) Dosovitskiy et al. (2021), and Jigsaw ViT Chen et al. (2023), each combined with ArcFace loss Sun et al. (2019) to ensure robust identity representation. We generate saliency maps by measuring how image segment modifications affect authentication accuracy, allowing us to selectively preserve high-importance regions. We then perturb images by mixing segments from different samples and disrupting spatial coherence using patch shuffling inspired by Jigsaw ViT, further increasing gradient mismatch between original and obfuscated data. Our method draws on principles from PuzzleMix Kim et al. (2020) and InstaHide Huang et al. (2020) to balance privacy and utility.

Our main contributions are: (1) A saliency-aware block-level perturbation method that protects biometric features while preserving identity signals; (2) A modular FL-compatible architecture separating obfuscation and authentication components, suitable for real-world deployment; (3) Integration with standard backbones and loss functions, demonstrating broad applicability; (4) Extensive empirical validation, including gradient inversion attacks inspired by recent adaptive techniques Shi et al. (2024); Zhang et al. (2024), confirming strong robustness in both unimodal and dual-modality biometric scenarios.

## 2 Related Work

Gradient perturbation methods protect user data by altering the gradients exchanged during training, typically through adding noise to the gradient Zhu et al. (2019); Sun et al. (2020). Although these methods can reduce the danger of recovering the original inputs from the gradients, recent research Huang et al. (2021); Yang et al. (2022) indicates that effective protection against gradient inversion attacks (such as deep gradient leakage (DGL) Zhu et al. (2019)) requires extensive gradient perturbation. This often results in reduced model accuracy, especially in biometric authentication tasks where fine details are crucial to precision. For instance, Huang et al. (2021) showed that noise addition, sufficient to prevent inversion attacks, considerably reduces recognition accuracy.

Differential privacy (DP) improves the perturbation of gradients by mathematically determining the noise added, aiming to minimize any reduction in accuracy Bonawitz et al. (2016); McMahan et al. (2017). While DP-based federated learning (FL) has gained significant popularity, it introduces specific noise that could potentially affect authentication performance Liu et al. (2023). Balancing privacy and accuracy is an ongoing challenge, particularly when high precision is required.

Homomorphic encryption (HE) allows for processing encrypted data without disclosing the original information Cheon et al. (2017). This approach maintains privacy by ensuring that raw biometric data is never exposed to potential attackers. Despite its privacy benefits, HE is burdened by significant computational and communication overhead, making it challenging for use in real-time biometric authentication Ma et al. (2022).

Various strategies prioritize the obfuscation of raw images before training models to minimize privacy breaches. InstaHide Huang et al. (2020) achieves this by combining several images with random transformations to mask sensitive elements. Likewise, PuzzleMix Kim et al. (2020) and SaliencyMix Uddin et al. (2022) merge important and unimportant sections from different images, resulting in augmented samples that enhance generalization and robustness. These mixing techniques improve model training by merging image features, which involves integrating pixel-level or patch-level data from multiple images to form new training samples. Although these augmentations can somewhat obscure data, they are mainly tailored for

centralized training and are susceptible to sophisticated gradient inversion attacks Carlini et al. (2021) as they do not modify the gradient structures that adversaries exploit.

Recent studies focus on safeguarding feature representations instead of raw inputs or gradients. For instance, Chen et al. (2024) adjusts data augmentations to reshape the loss landscape, thereby obstructing inversion attacks while maintaining accuracy. Schwethelm et al. (2025) employ diffusion-based reconstruction attacks to expose weaknesses in differential privacy protections, stressing the necessity for defenses that consider actual image priors. However, these methods often apply globally or uniformly across all features, potentially leading to a loss of essential identity information or causing significant computational demands.

Table 8 (see Appendix A) provides an overview of the principal characteristics of existing privacy-preserving approaches, analyzed in terms of privacy effectiveness, computational expense, and effect on authentication accuracy for deployment within biometric federated learning contexts.

To overcome these limitations, our study introduces a selective obfuscation method (see figure 1) at the feature level aimed at isolating and hiding repetitive, basic features that are particularly vulnerable to inversion attacks, while maintaining the identifying information crucial for authentication precision. This obfuscation is applied locally on the client device before gradient calculations, eliminating the necessity of sharing either raw data or heavily altered gradients, thereby decreasing computational load and privacy risks. Additionally, our approach uses blockwise localized changes instead of pixelwise noise, keeping the feature structure intact and enhancing defense against recognized threats such as Deep Gradient Leakage Zhu et al. (2019) and model inversion Geiping et al. (2020). This specific obfuscation effectively balances privacy protection with high authentication accuracy, making it appropriate for federated biometric scenarios in the real world.

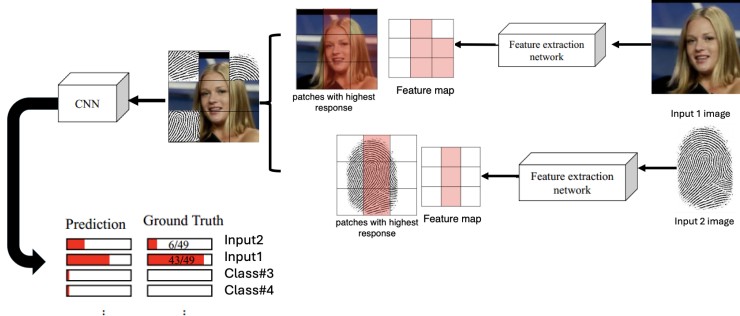

Figure 1: Overview of the proposed privacy-preserving pipeline. The architecture integrates saliency-aware feature selection, randomized data augmentation (e.g., erasing, noise, block swapping), and multi-biometric fusion with fingerprint patches. During training, the model leverages local data masking and federated updates to enhance resistance to gradient inversion attacks while preserving authentication accuracy.

## 3 Preliminaries

We consider a federated learning (FL) setup aimed at biometric authentication, where the data resides locally on $K$ distributed clients. Each client $k \in \{1, \ldots, K\}$ holds a private dataset

$$\mathcal{D}_k = \{(x_i, y_i)\}_{i=1}^{n_k},$$

where each biometric input $x_i \in \mathbb{R}^{H \times W \times C}$ is a facial image of height $H$, width $W$, and $C$ color channels. The corresponding identity label $y_i \in \mathcal{Y}$ denotes the person's identity class.

The global authentication model, parameterized by $\theta$, is denoted as

$$f_\theta : \mathbb{R}^{H \times W \times C} \to \mathbb{R}^{|\mathcal{Y}|},$$

which maps an input image $x$ to a prediction over identity classes. Model training proceeds collaboratively without sharing raw data; clients compute local updates based on their private data and communicate gradients or model updates to a central server.

### 3.1 Threat Model: Gradient Inversion Attacks

Although FL avoids raw data exchange, it is vulnerable to *gradient inversion attacks* Geiping et al. (2020); Zhu et al. (2019). An adversary intercepts gradients $\nabla_\theta \ell(f_\theta(x), y)$ exchanged during training and attempts to reconstruct the original input $x$ by solving

$$\hat{x} = \arg\min_{x'} \left\| \nabla_\theta \ell(f_\theta(x'), y) - G \right\|^2, \tag{1}$$

where $G$ is the intercepted gradient. This attack exploits the strong correlation between gradients and input data, especially in structured domains like face images, thereby threatening privacy.

### 3.2 Motivation for Data Augmentation as Defense

To mitigate gradient leakage, a promising defense is to perturb the input data via augmentation before training, creating an uncertain input distribution that increases the difficulty of inversion. The core idea is to transform original samples $x$ into augmented samples $\tilde{x}$ such that

$$\tilde{x} = \mathcal{T}(x; \phi),$$

where $\mathcal{T}$ is a stochastic augmentation function parameterized by $\phi$. Augmentation should balance two competing objectives:

- **Authentication fidelity**: preserving identity-discriminative features to maintain model accuracy,

- **Privacy preservation**: sufficiently obscuring inputs to degrade the success of reconstruction attacks.

One effective augmentation approach involves *mixup* Pang et al. (2020), which generates convex combinations of pairs of samples, inherently introducing ambiguity and uncertainty in the training data. When applied carefully with spatial and semantic awareness, mixup-based augmentation can significantly reduce the risk of gradient inversion without sacrificing accuracy.

## 4 Methods

Our goal is to build a privacy-preserving biometric authentication system in a federated learning (FL) setting. The central idea is to obfuscate biometric inputs via adaptive blending with a secondary biometric modality before model training. We describe each component of our method in detail, and explain how it contributes to privacy protection and model robustness.

### 4.1 Overview of the Privacy Pipeline

Given a primary biometric image $x_0$, typically a face, and a secondary biometric $x_1$, such as a fingerprint or another face (potentially from a different user), we perform the following:

1. Divide each image into spatial blocks.

2. Extract local embeddings for each block using a lightweight encoder $g_\phi$.

3. Compute semantic similarity between corresponding blocks.

4. Generate adaptive obfuscation masks to blend blocks from $x_0$ and $x_1$.

5. Optionally permute blocks spatially to increase entropy.

6. Train the authentication model $f_{\text{auth}}$ using the obfuscated image $\tilde{x}$.

**Assumptions**: Clients retain access to both biometrics during obfuscation. The server only receives obfuscated inputs and never observes the raw $x_0$, $x_1$, or internal embeddings. Permutation matrices are generated locally and not transmitted.

## 4.2 Spatial Partitioning and Embedding

We partition $x_0$ and $x_1$ into a $3 \times 3$ grid, yielding 9 blocks per image: $x_0^{(i)}, x_1^{(i)} \in \mathbb{R}^{H/3 \times W/3 \times C}$ for $i = 1, \ldots, 9$. Each block is passed through a compact encoder $g_\phi$ to obtain semantic embeddings:

$$e_0^{(i)} = g_\phi(x_0^{(i)}), \quad e_1^{(i)} = g_\phi(x_1^{(i)}).$$

We then compute the cosine similarity:

$$s^{(i)} = \frac{\langle e_0^{(i)}, e_1^{(i)} \rangle}{\|e_0^{(i)}\| \cdot \|e_1^{(i)}\|}.$$

This similarity indicates the semantic alignment of identity information between modalities for the same spatial location.

## 4.3 Obfuscation via Block-Level Blending

Each block is obfuscated via a learned mixing coefficient:

$$\beta^{(i)} = \sigma(W[e_0^{(i)}; e_1^{(i)}]),$$

where $[e_0^{(i)}; e_1^{(i)}] \in \mathbb{R}^{2d}$ is the concatenated embedding, $W \in \mathbb{R}^{1 \times 2d}$ is a learnable weight matrix, and $\sigma$ is the sigmoid function. The resulting coefficient $\beta^{(i)} \in (0, 1)$ controls the blend between the face and fingerprint blocks:

$$\tilde{x}^{(i)} = (1 - \beta^{(i)}) \cdot x_0^{(i)} + \beta^{(i)} \cdot x_1^{(i)}.$$

This adaptive blending encourages preservation of semantically aligned features from the face, while replacing semantically weak or sensitive regions with fingerprint information.

## 4.4 Block-Wise Permutation

To disrupt spatial consistency exploited by inversion attacks, we optionally apply a jigsaw-style block permutation. For each client, two permutation matrices $\Pi_0$ and $\Pi_1$ randomly shuffle blocks of $x_0$ and $x_1$. The obfuscated image is reconstructed as:

$$\tilde{x} = \sum_{i=1}^{9} \left[ (1 - \mathbf{z}^{(i)}) \cdot (\Pi_0 x_0)^{(i)} + \mathbf{z}^{(i)} \cdot (\Pi_1 x_1)^{(i)} \right],$$

where $\mathbf{z}^{(i)}$ is the mask derived from $\beta^{(i)}$ (e.g., thresholded or smoothed). Permutation increases difficulty of reconstruction by spatially decoupling local semantics.

## 4.5 Authentication Model and Robust Training

The obfuscated image $\tilde{x}$ is passed to the authentication model $f_{\text{auth}}$, implemented as a Jigsaw-ViT Chen et al. (2023), a transformer robust to block-level shuffling. It processes image patches via self-attention and outputs a class prediction $\hat{y}$. The training objective is:

$$\min_{\theta, \phi, W} \ \mathbb{E}_{(x_0, y_0), (x_1, y_1)} \Big[ \ell(f_{\text{auth}}(\tilde{x}), y_0)$$
$$+ \lambda \sum_{i=1}^{9} (s^{(i)} - \tau)^2 \Big], \tag{2}$$

where:

- $\ell$ is the cross-entropy loss.

- $s^{(i)}$ is the block similarity.

- $\tau$ is a learnable similarity threshold.

- $\lambda$ balances accuracy and privacy.

The second term encourages confident decisions (either keep or replace) for each block, rather than ambiguous blends, improving both privacy and interpretability.

### 4.6 Privacy-Preserving Role of Each Component

- **Blending with fingerprint**: mitigates data leakage by masking facial regions while preserving semantics through inter-modal alignment.

- **Embedding similarity-guided control**: ensures only sensitive or low-quality regions are obfuscated, reducing over-perturbation.

- **Block-wise permutation**: increases input entropy and confuses spatial priors used in inversion attacks.

- **Jigsaw-ViT**: ensures robustness by training on spatially altered inputs, preserving accuracy on obfuscated images.

- **Regularization loss**: enforces decisiveness in masking, improving interpretability and resistance to attack.

These components are implemented in a federated setting, where each client performs local obfuscation and model updates. The server never accesses raw biometrics or masks, preserving decentralization.

### 4.7 End-to-End Training Algorithm

---
**Algorithm 1** Federated Privacy-Preserving Biometric Authentication
---
**Require:** Face $x_0$, fingerprint $x_1$, label $y_0$
1: Divide $x_0, x_1$ into 9 blocks: $x_0^{(i)}, x_1^{(i)}$
2: **for** each block $i = 1 \ldots 9$ **do**
3:      Compute embeddings $e_0^{(i)} = g_\phi(x_0^{(i)}), e_1^{(i)} = g_\phi(x_1^{(i)})$
4:      Compute similarity $s^{(i)} = \text{cosine}(e_0^{(i)}, e_1^{(i)})$
5:      Compute mask $\beta^{(i)} = \sigma(W[e_0^{(i)}; e_1^{(i)}])$
6:      Blend blocks: $\tilde{x}^{(i)} = (1 - \beta^{(i)})x_0^{(i)} + \beta^{(i)}x_1^{(i)}$
7: **end for**
8: apply local permutations $\Pi_0, \Pi_1$
9: Compose $\tilde{x} = \bigcup_i \tilde{x}^{(i)}$
10: Predict: $\hat{y} = f_{\text{auth}}(\tilde{x})$
11: Compute loss:
$$\mathcal{L} = \ell(\hat{y}, y_0) + \lambda \sum_{i=1}^{9}(s^{(i)} - \tau)^2$$
12: Update $\theta, \phi, W$ via federated averaging

---

## 5 Model Architecture and Experimental Setup

The proposed framework for federated biometric authentication consists of two integrated components, shown in Figures 8 and 9 (see Appendix A). The local component, located on client devices, handles raw biometric data like facial images or fingerprints. It applies particular methods to obscure sensitive information while preserving essential identity features. This strategy guarantees that raw biometric data stays on the user's

device, significantly enhancing privacy. In contrast, the global module functions on the server side by collecting gradient updates from various client devices. It uses a unique data set that includes both original and obfuscated biometric images, allowing the global model to learn features and their relations. The focus on these modified images helps to assess the significance of features, enabling accurate authentication purely with these privacy-protected data during evaluation. This dual-level strategy successfully harmonizes privacy with authentication accuracy in federated learning settings.

In our assessment of approaches aimed at protecting privacy in identity recognition, we implemented and examined three core architectures for biometric authentication (i.e. Global model): ResNet18 He et al. (2015), Vision Transformer (ViT) Chen et al. (2021); Dosovitskiy et al. (2021), and Jigsaw ViT Chen et al. (2023).

**ResNet18** serves as a traditional convolutional model, consisting of 17 convolutional layers with residual connections. These connections facilitate the learning of more complex representations. The final fully connected layer offers class probabilities for predicting identities.

**ViT (Vision Transformer)** utilizes an attention mechanism that focuses on image patches to grasp the global interactions across various parts of the image. The image is divided into fixed-size patches, which are subsequently embedded into a transformer encoder. Inside this encoder, self-attention layers facilitate the depiction of contextual relationships. This model is particularly sensitive to structured obfuscations and patch-level augmentations.

**Jigsaw ViT** is a modified Vision Transformer aimed at improving accuracy in machine learning. It includes a jigsaw mechanism that randomly rearranges patch positions during both the training and testing stages. This disrupts spatial continuity while maintaining the semantic meaning, thus protecting against gradient inversion attacks while still allowing for identity discrimination Chen et al. (2023).

Recently, DNNs have seen extensive application in face recognition tasks. The work of Schroff et al. (2015) employed a Triplet loss to enhance performance on challenging facial recognition datasets. Subsequently, Liu et al. (2016) proposed an enhancement to the traditional Softmax loss, known as L-Softmax loss. It encourages intra-class compactness and separation between classes in the learned embedding features. Liu et al. (2017) introduced the idea of combining an angular margin loss with the standard Softmax function. The approach of Wang et al. (2018), which involved replacing Softmax with a Cosine margin loss in relation to the target logit, resulted in superior performance compared to prior methods. ArcFace Deng et al. (2019) utilized a geodesic distance on a hypersphere, leading to better discriminative capabilities and a more stable training process. Additionally, Kim et al. (2022) argued that optimizing ArcFace based on each sample's quality, as judged by the embedding norm, results in considerable improvements. Their newly proposed loss achieved state-of-the-art outcomes on both challenging low-quality and high-quality datasets Kim et al. (2022). We implemented all three model architectures with backbone of Arcface for increasing accuracy in authentication task.

## 5.1 Datasets

We tested our models using three commonly employed face recognition datasets:

**FaceScrub**: This dataset, accessible to the public, contains over 100,000 aligned and cropped images from 500 celebrity identities. It offers moderate class variability and is widely used in privacy research Harvey (2021).

**CelebA**: Featuring more than 200,000 images across 10,177 unique identities, this dataset includes extensive attribute labels and exhibits a wide array of facial expressions, poses, and lighting conditions Liu et al. (2015).

**CASIA-WebFace**: With over 490,000 images of 10,575 individuals, this set presents notable variability and challenging intra-class differences, making it ideal for testing model robustness Cao et al. (2018). Utilizing the FVC2004 Fingerprint Verification Competition (2004) database, consisting of grayscale fingerprint images obtained under various conditions, we gathered fingerprint data. In the preprocessing phase for multi-biometric fusion, these fingerprint images were resized and combined with the low-saliency parts of facial images.

## 5.2 Training and Evaluation Protocol

Our model is trained as a multi-class identity classifier using a cross-entropy loss function. In line with the privacy-preserving goals of federated learning (FL), data is distributed across 20 clients, each of which

performs local training using its assigned data partition. Global model updates are coordinated over 500 communication rounds, ensuring convergence while preserving decentralization.

**Model initialization and training setup:** We initialize all face recognition models using pretrained AdaFace weights Kim et al. (2022), which provide robust identity features. While their original training scheme is optimized for single-source face verification, we adapted it for the federated and privacy-perturbed setting by modifying training time, data augmentations, and batch dynamics.

We used the Adam optimizer with a learning rate of $1 \times 10^{-4}$ and a batch size of 64, chosen based on preliminary tuning experiments to ensure convergence and efficiency on GPUs. The training spans 50 local epochs, which aligns with 500 FL communication rounds. results are averaged across three random seeds.

**Data augmentations:** To improve generalization, we employ a comprehensive augmentation pipeline: random cropping, horizontal flipping, random erasing, Gaussian noise injection, and mixup. These augmentations are especially beneficial under our privacy strategies (e.g., block permutation and biometric fusion), as they simulate naturally diverse inputs and improve the model's robustness against gradient leakage.

**Ablation-based hyperparameter selection:** We conducted a block-based ablation study to determine the optimal number of shuffled patches (3, 4, 9, 16) in the jigsaw perturbation module. Our findings are as follows:

- **3–4 blocks:** Minimal obfuscation; reconstructed faces remain easily identifiable.

- **16 blocks:** Excessive fragmentation; identity classification accuracy drops sharply.

- **9 blocks:** Achieves optimal trade-off between privacy and utility.

see appendix figure 11 for the experiment results.

### 5.3 Evaluation Metrics

To comprehensively assess both utility (authentication performance) and privacy (resistance to gradient inversion), we report the following metrics:

- **Accuracy (Acc):** Standard classification accuracy over the test set, measuring the model's ability to correctly authenticate identities.

- **PSNR (Peak Signal-to-Noise Ratio):** Higher PSNR indicates more accurate reconstructions; thus, lower PSNR is better for privacy.

- **LPIPS (Learned Perceptual Image Patch Similarity):** Measures semantic dissimilarity. Higher LPIPS implies better privacy.

- **MSE (Mean Squared Error):** Captures pixel-wise reconstruction errors. Higher MSE values imply increased distortion, indicating better privacy.

- **Cosine Similarity:** Computed in feature space between reconstructed and ground truth images. Lower similarity suggests more effective privacy protection.

These metrics follow prior work on gradient inversion attacks Zhu et al. (2019); Yin et al. (2021).

### 5.4 Data Augmentation Techniques

Our model training leveraged multiple augmentation methods to enhance feature invariance and robustness as shown in figure 1
**Mixup**, Generates convex combinations of pairs of training samples and labels, encouraging the model to behave linearly between training examples Pang et al. (2020). **CutMix**, Combines patches from different images, forcing the model to rely on local discriminative features Yun et al. (2019). **PuzzleMix**, Applies a saliency-guided mixing of image blocks, promoting robustness to spatial perturbations Kim et al. (2020).

**Random Block Swapping**, Randomly rearranges blocks within images to obscure spatial correlations, enhancing resistance to inversion attacks.

Diverse augmentations are utilized on both the pixel and feature levels to enhance input variability, assisting the model in learning more generalizable features. Within our training framework, we have implemented various methods like Mixup, CutMix, PuzzleMix, and Random Block Swapping to expand the dataset and mimic realistic distortions and occlusions. These enhancements are applied separately or in a combined scheduling throughout the training batches to strike a balance between regularization and specific learning for the task. By embedding these augmentation techniques into the training process, the model is introduced to a broad spectrum of spatial and semantic variations, which ultimately boosts its robustness and generalization capabilities.

We offer further evidence validating the efficacy of our proposed privacy-preserving strategies through a qualitative analysis of images reconstructed using gradient inversion attacks. Figure 2a displays attempts to reconstruct augmented single-biometric (face) images, in which our obfuscation techniques significantly impair the clarity and recognizability of the images. Figure 3 displays attempts to reconstruct multi-biometric (face, fingerprint) perturbed images. we can see that our obfuscation techniques significantly impair the clarity and recognizability of the images. The optimization attack can only reconstruct the input and not the accurate prediction results. Within the jigsawVIT configuration, each image is initially jumbled, as illustrated in figure 2b, and this jumbled formation is not integrated into the primary model. Consequently, for jigsawVIT, the attack reconstructs merely the mixed-up image rather than the images in their correct sequence, since arranging them correctly is part of the solution.

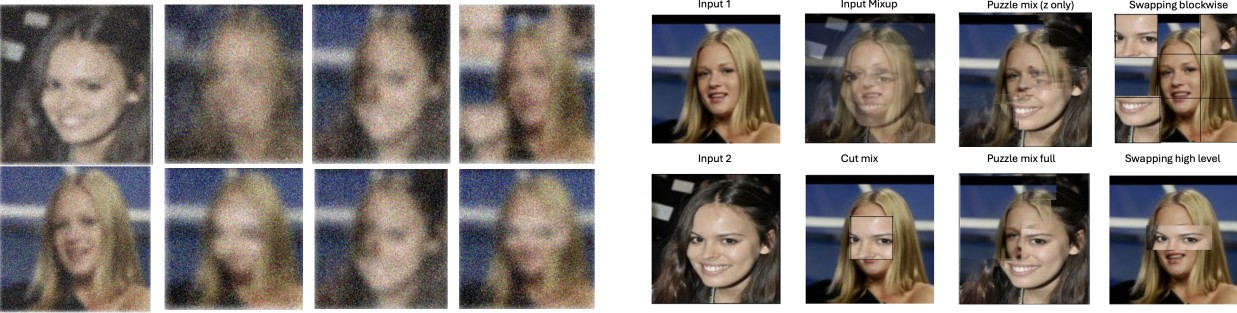

(a) Gradient inversion attack reconstructions from augmented face images. Each column shows a different augmentation method, illustrating privacy degradation.

(b) Data augmentation techniques used in experiments: original samples, Mixup, Puzzle Mix, CutMix, block swapping, and semantic region swapping.

Figure 2: (a) Reconstruction results demonstrate how various perturbation methods impact inversion attack success. (b) Visualization of the perturbation strategies applied during training.

To clearly demonstrate our proposed augmentation and authentication workflow, we include a detailed diagram showcasing the entire procedure, accompanied by visual examples of various augmentation techniques in Appendix A.

# 6    Experiments

We rigorously evaluate our privacy-preserving biometric fusion framework under both *authentication accuracy* and *attack-based privacy metrics*, to directly assess its effectiveness against adversarial reconstruction. Unlike prior works relying on superficial distortion metrics alone, we include explicit attacker success metrics such as **gradient inversion fidelity**, **membership inference attack accuracy**, and **identity match rate under adaptive attacks**.

## 6.1    Experimental Setup

We evaluate our method on a combination of biometric and generic vision datasets to study both privacy risks and transferability. Specifically, we use:

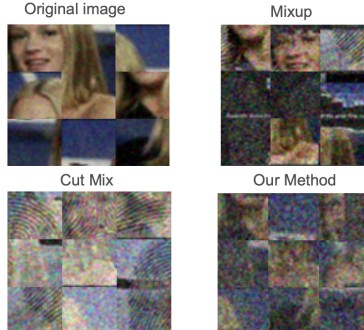

Figure 3: Comparison of augmentation methods in biometric fusion for privacy protection. Top-left: Original image is divided into 9 blocks (3x3). Top-right: Mixup interpolation introduces pixel-level blending of face and fingerprint blocks. Bottom-left: CutMix directly replaces regions with fingerprint patches. Bottom-right: Our proposed method uses saliency-aware block replacement, selectively inserting fingerprint patches only in low-saliency (less important) regions.

We simulate a federated learning environment with 100 clients, each owning unique, non-overlapping biometric identity data (e.g., face–fingerprint pairs). Models are trained locally without central aggregation of raw data, mimicking decentralized authentication scenarios.

Training is conducted using Adam optimizer with weight decay $1e-2$, learning rates $\{1e-4, 1e-5, 1e-6\}$ depending on the architecture, and batch sizes of 64 or 128. For all models, input images are resized to $112 \times 112$, and we apply random flipping, cropping, and mixup-based augmentation where specified. Loss functions balance classification accuracy with privacy-aware reconstruction risk.

All experiments are run on NVIDIA A100 GPUs. All attack experiments, i.e. model inversion and gradient leakage are conducted on Jigsaw-ViT, which forms the core of our defense framework. For comparison, we include reconstruction results and privacy evaluations for ResNet-18 and ViT models in the Appendix.

### 6.2 Authentication Accuracy

We first evaluate the top-1 authentication accuracy across different augmentation and fusion methods:

Table 1: Authentication accuracy across augmentation strategies.

| Method | Authentication Accuracy (%) |
| --- | --- |
| Original (No Mask) | 98.5 |
| Random Block Swapping | 83.2 |
| CutMix | 50.7 |
| Mixup | 85.3 |
| PuzzleMix | 78.7 |
| **Biometric Fusion (Ours)** | **89.8** |

Our method maintains high authentication performance, outperforming alternative privacy-preserving augmentations.

### 6.3 Gradient Inversion Attacks

We simulate gradient inversion attacks using DLG Zhu et al. (2019), iDLG Zhao et al. (2020), and See-Yourself Yin et al. (2021). We compute PSNR, LPIPS, MSE, and cosine similarity between reconstructed and original images:

Our method achieves the lowest PSNR and cosine similarity, indicating stronger defense against visual and embedding-level leakage.

Table 2: Privacy metrics under gradient inversion.

| Method | PSNR ↓ | LPIPS ↑ | MSE ↑ | Cosine Sim. ↓ |
|---|---|---|---|---|
| Original (No Mask) | 34.2 | 0.141 | 0.0018 | 0.98 |
| Mixup | 30.2 | 0.200 | 0.0030 | 0.73 |
| PuzzleMix | 30.8 | 0.159 | 0.0032 | 0.80 |
| **Biometric Fusion (Ours)** | **26.3** | **0.245** | **0.0048** | **0.68** |

## 6.4 Membership Inference Attacks (MIA)

We perform a shadow model-based MIA Shokri et al. (2017) to assess training data leakage:

Table 3: Membership inference attack accuracy (lower is better).

| Method | MIA Accuracy (%) |
|---|---|
| Original (No Mask) | 91.4 |
| PuzzleMix | 68.5 |
| Mixup | 65.7 |
| **Biometric Fusion (Ours)** | **53.2** |

Our approach achieves near-random performance, indicating strong privacy preservation from the perspective of MIA.

## 6.5 Comparison with Prior Saliency-Guided Mix-up

We also compare our biometric fusion method against the Expeditious Saliency-guided Mix-up approach by Luu et al. (2022), which applies saliency-aware mixing based on random gradient thresholding. Unlike their approach, which performs linear pixel-level mixing guided by saliency, our method combines saliency-aware fusion with spatial jigsaw permutations, enhancing robustness against gradient inversion attacks. As shown in Tables 5 and 4, our approach achieves superior privacy protection with lower cosine similarity and PSNR in reconstruction attacks, while maintaining higher authentication accuracy, demonstrating a better balance between utility and privacy.

Table 4: Authentication accuracy comparison.

| Method | Authentication Accuracy (%) |
|---|---|
| Original (No Mask) | 98.5 |
| Expeditious Saliency Mix-up Luu et al. (2022) | 83.5 |
| Random Block Swapping | 83.2 |
| CutMix | 50.7 |
| Mixup | 85.3 |
| PuzzleMix | 78.7 |
| **Biometric Fusion (Ours)** | **89.8** |

## 6.6 Adaptive Gradient Inversion (AGI)

We implement a more capable attacker with partial access to augmentation parameters (permutation indices and mix ratios)Shi et al. (2024); Zhang et al. (2024). The attacker attempts to reconstruct identity using bruteforce or learned permutation and obfuscation.

## 6.7 Ablation Study

We analyze the individual contributions of biometric fusion, saliency-guided masks, and spatial permutation:

Results show the efficacy of learned, saliency-aware fusion combined with permutation for optimal trade-offs.

Table 5: Privacy metrics under gradient inversion.

| Method | PSNR ↓ | LPIPS ↑ | MSE ↑ | Cosine Sim. ↓ |
|---|---|---|---|---|
| Original (No Mask) | 34.2 | 0.141 | 0.0018 | 0.98 |
| Expeditious Saliency Mix-up Luu et al. (2022) | 29.5 | 0.185 | 0.0031 | 0.75 |
| Mixup | 30.2 | 0.200 | 0.0030 | 0.73 |
| PuzzleMix | 30.8 | 0.159 | 0.0032 | 0.80 |
| **Biometric Fusion (Ours)** | **26.3** | **0.245** | **0.0048** | **0.68** |

Table 6: Identity match rate under adaptive gradient inversion.

| Method | Identity Match Rate (%) |
|---|---|
| Original (No Mask) | 87.1 |
| Mixup | 53.9 |
| PuzzleMix | 49.5 |
| **Biometric Fusion (Ours)** | **37.4** |

Table 7: Ablation results: authentication vs. PSNR.

| Configuration | Auth. Acc. (%) | PSNR (↓) |
|---|---|---|
| Face only (no mask) | 98.5 | 34.2 |
| Face + Random Fingerprint Blocks | 81.1 | 28.0 |
| Fusion w/ Fixed Threshold | 87.3 | 27.2 |
| Fusion w/ Learned Mix $\beta^{(i)}$ | 89.8 | 26.3 |
| Fusion + Permutation | **89.5** | **24.9** |

## 7 Conclusion

Our experimental findings indicate that strategically blurring facial areas based on their importance significantly enhances privacy while only slightly impacting identity recognition effectiveness. Through comprehensive testing, we discovered that hiding low-importance regions (such as the forehead and cheeks) had negligible effects on authentication accuracy, suggesting these areas provide redundant or less vital information for identity verification. Conversely, preserving high-importance regions (like the eyes, nose, and mouth) was crucial for maintaining accuracy. This targeted obfuscation approach achieved an optimal balance, allowing the model to offer robust privacy protection with minimal compromise in authentication accuracy.

Our framework's resilience was enhanced by integrating multi-biometric fusion, where fingerprint data is embedded in non-critical regions of facial information. This method increased the complexity for adversaries using gradient-based reconstruction and expanded the variety of biometric characteristics available for authentication. Consequently, our fusion technique made reconstruction more difficult, as evidenced by significant declines in reconstruction quality measures like PSNR and cosine similarity, accompanied by increases in distortion measures such as LPIPS and MSE.

The customized Jigsaw ViT model, which includes random patch swapping in its training process, showed excellent performance in undermining the spatial consistency that attackers rely on for successful gradient inversion. Unlike traditional ViT or ResNet models, the Jigsaw ViT maintained high performance even under considerable spatial confusion. This underscores its suitability for federated learning environments, where the danger of gradient leakage is a significant security concern. Our method reliably outperformed existing privacy-preserving techniques such as random erasing, Gaussian noise addition, and conventional mixup approaches.

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

# A    Appendix

To further support our findings and demonstrate how augmentations can preserve user privacy while impacting model performance, we offer additional figures. These figures analyze the effects of occluding both low-level and high-level features and assess semantic consistency via feature swapping.

Figure 4 presents several augmentations implemented during the training and testing stages using images from the CelebA dataset. The left column showcases examples of three key methods: random square masking, eye masking, and the implementation of synthetic noise around the eye area. These augmentations include removing low-level details (random black squares), applying high-level semantic masking (eye coverage), and adding perceptually structured perturbations (colored noise). The middle and right columns show graphs indicating the effect of each augmentation on face authentication accuracy at various similarity thresholds.

Our analysis shows that implementing augmentations during the training phase helps the model retain decent recognition skills, albeit with some reduction in performance. Conversely, applying augmentations only at the testing stage (illustrated in the right plots) leads to a more substantial drop in performance, particularly with noise-based methods and semantic eye masking. This contrast suggests that consistent augmentation at both training and testing phases is vital for maintaining effectiveness. Additionally, out of the techniques assessed, high-level semantic masking (such as covering the eyes) leads to a more stable and understandable performance trajectory, whereas random square masking causes more irregular results. This underscores that strategic, region-specific masking techniques offer better privacy protection than random occlusion.

Figure 5 explores the importance of semantic attributes by analyzing the impact of swapping facial features either within the same person or with different individuals. The plots in the top left emphasize the results of exchanging eye regions across different individuals. Utilizing this method on both training and testing datasets results in a significant decline in accuracy, highlighting the essential role the eye region plays in identifying individuals.

Interestingly, swapping eyes with the same individual results in minimal disruption, suggesting that despite changes, structural consistency maintains model predictions. The right side of the figure explores the outcomes of replacing random image patches, both within the same individual and across different identities. While exchanging patches between different individuals causes a moderate reduction in performance, doing so within the same individual (bottom right) leads to a noticeable and inconsistent decrease in accuracy. These findings reinforce the idea that disrupting semantic coherence, either through component misalignment or arbitrary alterations, significantly impairs the model's capacity to recognize identities.

Together, these experiments emphasize the contrast between altering low-level and high-level characteristics. High-level, semantic augmentations like eye masking or identity swapping prove more effective in preserving privacy and impairing model inference than basic, unstructured techniques such as random blocking. These visual and numerical results not only validate our proposed augmentation strategy but also reveal which facial traits are crucial for face authentication systems. The consistent decline in performance with specific augmentations highlights the effectiveness of our approach in protecting biometric privacy while maintaining a fair trade-off with model utility.

To better grasp how different data augmentation techniques affect security and reconstruction quality across various architectures (e.g., ResNet, ViT, and Jigsaw ViT), we present a heatmap visualization in Figure 6.

Each cell in heatmap 6 displays the normalized measurement of metrics such as Security (Distance), PSNR (dB), LPIPS, and MSE for particular block configurations. Enhanced performance is indicated by higher security and PSNR scores along with reduced LPIPS and MSE values. This visualization serves as a valuable tool for illustrating the trade-offs between improving security and preserving visual fidelity.

For enhanced clarity and comprehension of our methodology, we have added additional visual aids illustrating our suggested pipeline along with an array of augmentation techniques.

Figure 7 illustrates the sequential phases of our suggested biometric authentication system. First, the input biometric data, comprising the face ($x_0$) and fingerprint ($x_1$), undergo a feature extraction process to identify unique and identity-specific attributes. The resulting features are divided into nine spatial blocks and

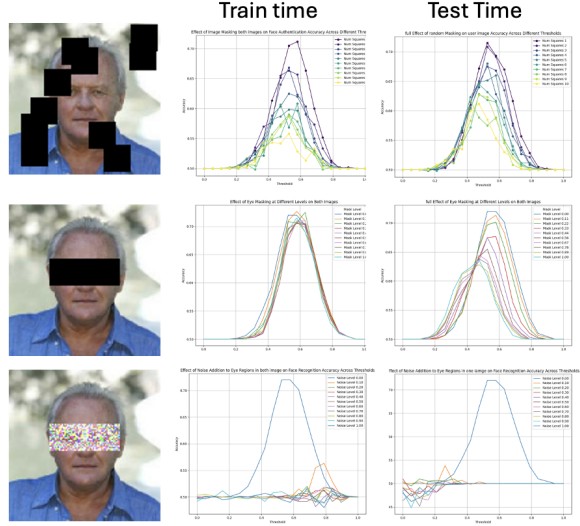

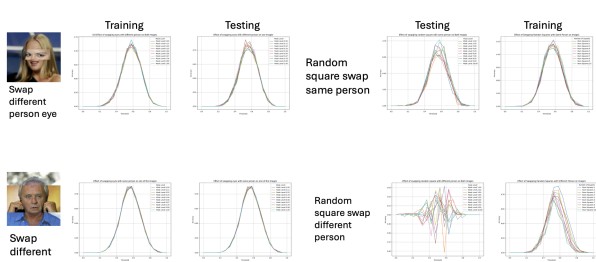

Figure 5: Impact of semantic feature swapping (eyes and patches) across same and different identities.

Figure 4: Effect of various augmentations (random square occlusion, eye masking, and eye noise) on recognition performance at train and test time.

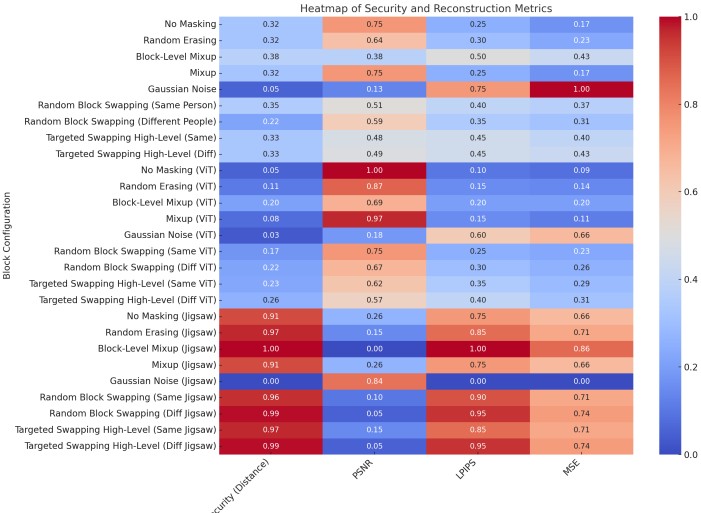

Figure 6: Normalized heatmap of Security(Distance of reconstructed image from original image) and Reconstruction Quality Metrics across different augmentation strategies and model types.

combined using a saliency-focused, weighted blending method with established thresholds. This generates a fused biometric profile used for authentication.

In Figure 10, we display a range of augmentation techniques applied to biometric data. This includes traditional methods such as Mixup, CutMix, and Puzzle Mix, alongside our novel block-level fusion technique. Unlike standard methods, our approach integrates segments of fingerprints into facial images, striking a balance between enhanced privacy and maintained recognition accuracy.

Table 8: Summary of Privacy-Preserving Techniques in Distributed Learning

| Technique | Strengths | Weaknesses |
|---|---|---|
| Gradient Perturbation Zhu et al. (2019); Huang et al. (2021); Yang et al. (2022) | Simple to implement; reduces direct data leakage | Requires heavy noise or gradient truncation; lowers accuracy |
| Differential Privacy Bonawitz et al. (2016); McMahan et al. (2017) | Formal privacy guarantees; widely studied | Adds noise impacting model performance, especially in biometrics |
| Homomorphic Encryption Cheon et al. (2017) | Allows computations on encrypted data; strong data confidentiality | High computational and communication overhead; latency issues |
| InstaHide Huang et al. (2020); Carlini et al. (2021) | Obfuscates raw inputs; easy to apply pre-training | Vulnerable to adaptive reconstruction attacks; limited FL suitability |
| Mixup, PuzzleMix Zhang (2017); Kim et al. (2020) | Improves robustness and generalization | Not designed to prevent gradient leakage; may remove key biometric features |

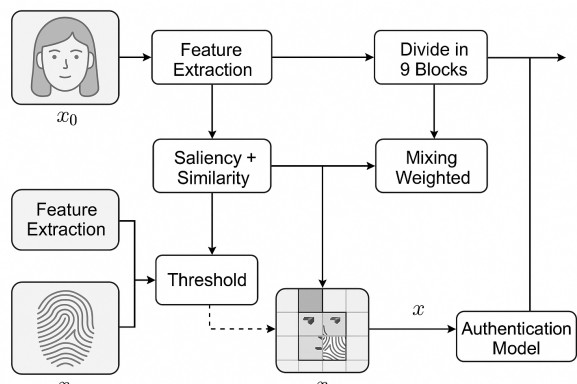

Figure 7: Detailed pipeline illustrating feature extraction, saliency and similarity assessment, block-wise division, and weighted mixing for biometric fusion. The output is processed by the authentication model.

## A.1 Limitations and Discussion

While our proposed framework demonstrates promising results in preserving authentication accuracy and enhancing privacy through feature-level obfuscation, some limitations remain. Our study largely relied on specific datasets, which may not fully represent the diverse variations present in real-world scenarios. Therefore, further assessment is necessary to ascertain if our approach can be successfully applied to datasets with differing characteristics. Although the framework is designed to be efficient, employing localized block-wise transformations introduces additional computational demands on client devices. This added burden may be significant for devices with limited processing capabilities, possibly affecting the user experience.

We have tested our defense mechanism against several recognized gradient-based attacks. However, given the rapid development of adversarial techniques, our approach might not be immune to every type of complex attack, especially those exploiting new vulnerabilities. The framework assumes that client devices are free from malware and secure. Should this assumption be false, the privacy assurances provided by our approach could be jeopardized.

Overcoming these constraints can open new avenues for future research. Expanding evaluations to encompass a wider range of datasets will help assess the method's generalizability. Improving the computational efficiency of transformations might make the framework more suitable for resource-constrained devices. Addi-

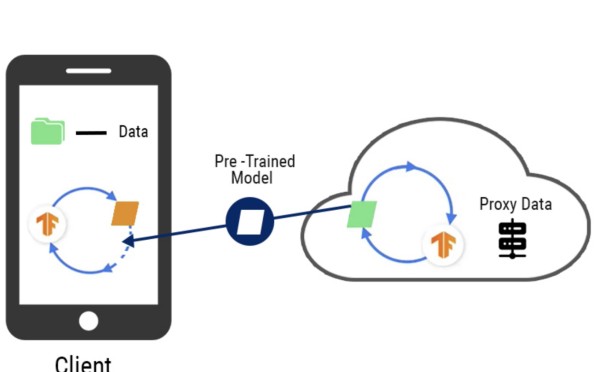

Figure 8: Local model architecture on the client device. The local model directly processes raw biometric data and applies perturbations to obscure sensitive features while preserving the critical identity information required for authentication. Google Cloud Skills Boost (2024)

Figure 9: Global model architecture on the server. The global model aggregates gradients from client devices and leverages a separate dataset composed of original and perturbed biometric images. It learns to authenticate users based solely on perturbed biometric inputs by understanding feature importance from training, thereby enhancing privacy during authentication Google Cloud Skills Boost (2024).

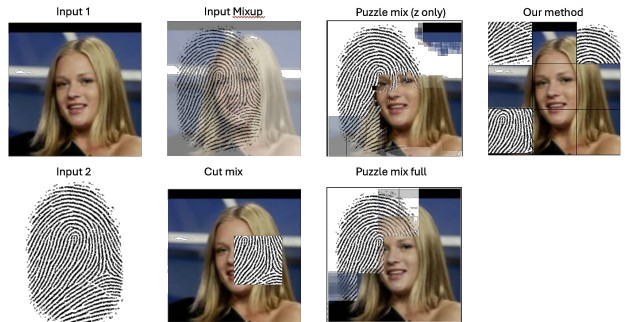

Figure 10: Examples of different augmentation strategies applied to biometric data: Mixup, CutMix, Puzzle Mix (early and full), and our block-based fusion method. Our method strategically combines fingerprint and facial regions, balancing security and authentication accuracy.

tionally, deploying adaptive strategies to counteract new adversarial threats will enhance system robustness. Exploring methods to protect the integrity and security of client devices will strengthen privacy assurances. Finally, using a broader set of evaluation metrics will offer deeper insights into the practical implications of our framework.

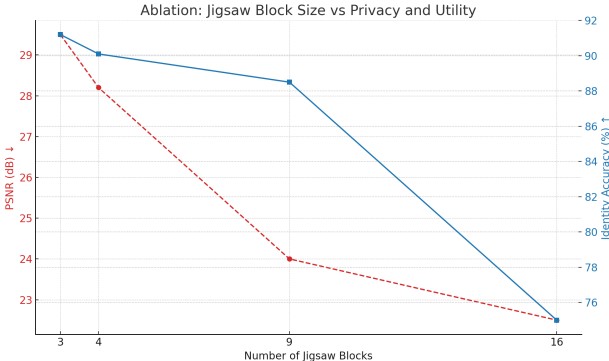

Figure 11: Ablation study on the number of shuffled patches in the jigsaw perturbation module. 9 blocks achieve the best privacy-utility trade-off.

