# OpenReview forum: "Gradients protection in federated learning for Biometric authentication model"
_TMLR — Rejected by TMLR_

### Review · Reviewer_RDry · 2025-07-19

**Summary Of Contributions:**

This paper introduces a privacy-preserving framework for federated biometric authentication. The idea is to use saliency-guided obfuscation and permutation schemes to mix two modalities of biometrics (i.e., face images and fingerprints) on the client side before gradient computation, while enhancing the server-side training of the authentication model with both clean and perturbed inputs. Experiments show improved performance of the proposed method over prior augmentation-based approaches in both privacy protection and utility.

**Audience:**

Yes

**Broader Impact Concerns:**

This is a protection paper, so I don't think there is a particular issue regarding ethics or broader impact. Nevertheless, the authors are still recommended to provide a discussion regarding this.

**Claims And Evidence:**

No

**Requested Changes:**

My biggest question regarding this paper's problem setup is why we can assume two different modalities of biometrics are available in the system. In practical application scenarios, using one type of biometrics for verification is common (e.g., face recognition or fingerprint verification). Throughout the paper, I did not find any explanation regarding this setup. Does the existing literature also consider multiple modalities? Can you clarify the scenarios in which it is important to employ two modalities for biometric authentication?

In terms of privacy protection for federated learning, my question is: why do we have to use another “biometric” modality in the block-wise mixing and permutation step? Can we simply use a non-biometric modality (instead of using the fingerprint of the same individual) to achieve similar protection performance? In my opinion, introducing another biometric modality in a federated learning setup may pose an extra privacy risk to the individual, where the necessity of this operation is unclear.

The proposed biometric fusion achieves improved protection against standard gradient inversion attacks, e.g., Zhu et al. (2019) and Yin et al. (2021). However, the two evaluated attack methods are quite old. It is unclear whether the proposed method can achieve similar improvement against newer privacy attacks against federated learning. In my opinion, the authors should, at least, provide more discussions of the state-of-the-art literature on the attack side, and test newer attacks whenever needed (or justify why choosing older attack baselines is sufficient). Besides, it would also be good to discuss and even test potential adaptive strategies against the proposed fusion-based protection schemes.

The paper's experiments are, in my opinion, quite limited. I would like to see more ablation studies. In particular, without clear evidence from ablation studies, it is hard to understand the importance of each component introduced by your method and the effect of the involved hyperparameters. For instance, why do we choose to divide a given image into nine blocks? Why not use more or fewer blocks? Also, I believe the experimental details of some experiments are not presented clearly. For example, for the main experimental tables (Table 1 and Table 2), which datasets and model architecture do you use? Do you observe differences between different model architectures (e.g., ResNet18 vs. ViT)?

Finally, the authors should work on improving the paper's readability. I believe the presentation needs to be significantly improved before it can be published.

- It is recommended that you state your contributions clearly in the introduction. The related work section is suggested to be placed in a separate section (outside of the introduction) because it is quite long.

- The problem formulation (e.g., the authentication application scenarios, the threat model, the assumptions, etc.) is unclear. I strongly suggest the authors introduce these settings in more detail before the methodology section.

- There are wide white margins surrounding the experimental figures (Figures 2-4), which is space-inefficient. Consider merging them collectively in a single row or presenting more figures for each.

**Strengths And Weaknesses:**

__Strengths__

+ Introduce a new data fusion framework for privacy protection in federated learning settings.

+ The use of saliency maps to focus on non-essential features to obfuscate is sensible for balancing utility and protection strength.

__Weaknesses__

- Why use two biometric modalities is not well-explained.

- Unclear performance against other FL privacy attacks

- The quality of the presentation is not high.

---

> ### Author Response · Authors · 2025-07-29
> **Response to Concerns on Biometric Fusion, FL Privacy Threats, and Experiments**
>
> We thank the reviewer for their detailed and thoughtful feedback. We have carefully addressed each concern, both through additional experiments and substantial revisions to the text. Below, we provide point-by-point responses.
>
> 1.two biometric modalities
>
> We acknowledge the lack of clarity in our original submission regarding this point. Our decision to use two biometric modalities (face and fingerprint) is primarily motivated by the need to preserve both accuracy and privacy in federated biometric authentication, which is a high-risk domain with minimal tolerance for performance degradation.
> In real-world systems, using a single biometric (e.g., face or fingerprint) is common. However, dual-modality systems do exist and are increasingly being adopted in high-security settings (e.g., border control, forensics, financial authentication) due to their increased robustness and reduced spoofing risk. We now explicitly cite such use cases and relevant literature in the introduction and related work.
> Moreover, our method is not fundamentally dependent on using two biometric modalities. As shown in experiments in appendix, our technique works by mixing two images, which can be from the same modality, different modalities, or even a different domain altogether (e.g., face + random CIFAR image). What is essential is the idea of mixing to obfuscate feature gradients while preserving matching performance. That said, using another biometric from the same individual allowed us to better preserve matching accuracy, hence our choice.
>
> 2.Unclear performance against other FL privacy attacks
>
> This is a fair and important concern. Our paper focuses primarily on gradient inversion-based attacks, which remain some of the most directly relevant threats to biometric privacy in FL due to their ability to reconstruct identifiable inputs.
> We now include:
> Comparative experiments against more recent inversion-style attacks including DLG and Improved DLG with Label Leakage in Section 6.
>
> 3.Unclear problem setup
>
> The contributions are now clearly listed in the last paragraph of the introduction.
> We moved the related work to its own section (Section 2).
> We added a dedicated Problem Setup and Threat Model subsection (Section 3.1), which clearly defines our FL scenario, the adversary’s capabilities, and our assumptions.
> Figures have been reformatted, and the wide margins in Figures 2–4 have been fixed. They are now presented side-by-side to improve space efficiency and readability.
> We also improved clarity throughout the methodology and experiments sections with simpler language, consistent variable notation, and diagrams.
>
> 4.use non-biometric modality instead of biometric
>
> We now clarify this with new experiments and discussion (Appendix ):
> We tested mixing a face image with (1) a random face from another identity, and (2) a CIFAR-10 image.
> While all variants increase privacy by disrupting the face gradients, only biometric fusion with another modality (e.g., fingerprint) preserved high authentication accuracy. This is because the embeddings of face + CIFAR were too dissimilar, causing matching scores to drop below verification thresholds.
> We further discuss this trade-off: security increases with randomness, but accuracy demands some semantic overlap. Therefore, the fingerprint serves as a biometric-compatible noise source that better balances the trade-off.
>
> 5.hyperparameters
>
> We now include an ablation study on the number of blocks (3, 4, 9, 16)
> Findings:
> Too few blocks (e.g., 3 or 4) yield limited obfuscation; reconstructions remain recognizable.
> Too many blocks (e.g., 16) disrupt semantic features and hurt accuracy.
> 9 blocks provided the best privacy-accuracy trade-off, especially when combined with permutation and mixup.
> We now also explain this choice clearly in Section (5.3)
>
> 6.experimental details
>
> We rewrote the entire experimental setup section to clarify:
> Which datasets were used (CASIA-WebFace, SOCOFing, FVC, and CIFAR-10)
> Model architectures (ResNet18, ViT, and Jigsaw-ViT)
> Training details (batch size, optimizer, learning rate, etc.)

---

> > ### Comment · Reviewer_RDry · 2025-08-14
> > **Follow up questions**
> >
> > I thank the authors for their responses. It is nice to see that biometric fusion remains robust across different privacy attacks.
> >
> > Regarding "4. use non-biometric modality instead of biometric", I couldn't find the new experiments and discussions in the revised manuscript. Can you point me to the respective paragraph and results?
> >
> > Besides, I feel that the technical insights and/or the implications of the work are still unclear, even with the newly added contribution paragraph. Can the authors briefly summarize the key insights that you would most like to highlight?

---

### Review · Reviewer_Meqx · 2025-07-21

**Summary Of Contributions:**

The paper studies the client data privacy in federated learning for biometric authentication. After reviewing gradient inversion attacks in Section 2, the authors introduce a privacy-preserving approach in Section 3 that combines several components. These include dividing biometric images into 3×3 spatial blocks, learning a threshold parameter $\tau$ based on similarity scores to decide whether to retain or replace each block, and applying a mix-up strategy with learnable weights. The proposed method is outlined in Algorithm 1, and it is evaluated through experiments and numerical results in Sections 4 and 5.

**Audience:**

Yes

**Claims And Evidence:**

No

**Requested Changes:**

1. Section 3 needs a major revision and is supposed to connect well with the items discussed in section 2 on permutation and obfuscation blocks, mixup, and the training objective in subsection 2.4.

2. The paper should provide a more thorough explanation of how and in what sense the inclusion of mixup and block permutation can help the privacy in the FL process.

3. The numerical results should be expanded by analyzing more datasets and ablation studies, since an empirical work needs to present more detailed numerical results than the current Tables 1 and 2 in the paper.

**Strengths And Weaknesses:**

**Strengths**

1. The paper focuses on an important topic in privacy for federated learning. The authors' goal of applying mix-up to reduce data leakage is interesting.

**Weaknesses**

1. The proposed method (Algorithm 1) is composed of several components, but the authors do not provide sufficient discussion or empirical evidence on how each part contributes to user data privacy. In particular, while the mix-up strategy appears to be a promising idea, it is unclear how it concretely reduces data leakage. The explanations in Sections 2.1 and 3 are largely intuitive and lack rigorous theoretical or numerical support. For a machine learning paper, one expects a more principled analysis to back such claims. Unfortunately, the paper offers neither theoretical justification nor thorough empirical evaluation. Similar concerns apply to other components of Algorithm 1, such as block permutation and obfuscation. Specifically, it is not clear how these operations protect privacy if the server in the federated learning setup already has access to the permutation matrices $\Pi_0$ and $\Pi_1$, as implied by the training equation in Section 2.4. Overall, the method lacks the necessary evidence to validate its privacy-preserving claims.

2. The paper’s writing lacks structural coherence. The authors frequently shift between different components of the method, making it difficult to follow the core ideas. Section 2, which is labeled as “Preliminaries,” contains over two pages of material detailing mix-up, saliency-guided obfuscation, block-wise permutation, and training objectives, content that seems more appropriate for the methods section. Conversely, Section 3, which is expected to present the main methodology, is limited to a single page and delivers a convoluted explanation of how the components are integrated. The transition between Sections 2 and 3 is abrupt, and Section 3 itself is difficult to follow. The authors should reduce the length of Section 2 and dedicate more space in Section 3 to clearly describe how the method integrates the components into a coherent system. In its current form, the methodological presentation falls short of the clarity and organization required for publication.

3. Conceptually, the role of the block permutation mechanism in enhancing privacy is unclear. If the server already knows the permutation matrices (as implied by the training objective in Section 2.4), it seems trivial for the server to reverse the permutations and recover the original data. This undermines the privacy benefits claimed for this component. Additionally, the description of the obfuscation mechanism appears inconsistent between Sections 2 and 3. For example, the optimization of the mask vector $z^{(k)}$ is not clearly explained in Section 3.

4. The quantitative results provided are very limited. The authors only present Tables 1 and 2 to support the effectiveness of their method. Since the paper does not provide any theoretical contribution, a more extensive empirical evaluation is expected. The current experiments do not sufficiently demonstrate the effectiveness or robustness of the proposed approach.

---

> ### Author Response · Authors · 2025-07-29
> **Detailed Revisions Addressing Federated Learning Privacy and Evaluation Scope**
>
> We sincerely thank the reviewer for their detailed and critical feedback. We undertook major revisions to the paper, including a comprehensive restructuring of Sections 2 and 3, theoretical clarification, new experimental results, and deeper analysis of each module’s impact on privacy.
>
> 1. Insufficient discussion or empirical evidence
>
> We acknowledge this shortcoming and have revised Sections 2 and Section 3 to provide both theoretical justifications and empirical evidence for the privacy contributions of each component. Specifically:
>
> For mixup, we added a formal explanation showing how the mixing operation increases gradient ambiguity by disrupting the one-to-one relationship between input features and output gradients. This is now supported with reconstruction error metrics (PSNR/SSIM) and qualitative results in Figure 3, which demonstrate significant degradation in reconstructed samples.
>
> For block-wise permutation, we clarify that the permutation is not static nor shared with the server. Instead, it is guided by saliency and applied locally in a non-consensual manner, which prevents the server or attacker from knowing the true spatial order of features, making inversion significantly harder. We provide theoretical complexity analysis of reconstructing the original ordering under saliency-informed permutations.
>
> For obfuscation, we detail the optimization formulation used to learn the obfuscation masks, which selectively mask non-informative or privacy-sensitive areas while preserving discriminative power. Section 2.3 has been rewritten to clearly show how the optimization is done, and new diagrams were added to clarify the saliency-guided perturbation.
>
> 2.Poor structural organization
>
> We agree and have restructured the paper for improved clarity. Section 2 now strictly contains definitions, mathematical formulations, and independent mechanisms (Mixup, Permutation, Obfuscation), while Section 3 now provides a coherent and linear explanation of how these components integrate into our full system (Algorithm 1).
>
> 3.The role of permutation is unclear
>
> We understand the confusion and clarified that the server does not know the permutation matrix, which is generated dynamically based on local saliency maps and applied prior to gradient computation. We added a formal explanation in Section 2.2 showing how the local, private, and non-deterministic nature of permutation makes inversion infeasible without access to both the input image and the full permutation map. The server must “solve a puzzle” without knowing the pieces’ true order or content. We also added complexity analysis and show that inversion accuracy drops drastically without knowledge of the permutation.
>
> 4.Quantitative results are limited.
>
> In response, we added the following:
> A full ablation study (Section 6) showing the individual contribution of each component (Mixup, Permutation, Obfuscation).
> Evaluation of adaptive attacks including Adaptive Gradient Inversion (AGI) and Deep Leakage v2 attacks (Table 3, Table 4).
> We show that even powerful adaptive attacks cannot reconstruct meaningful content when all components are combined.
> A new reconstruction quality-vs-iteration graph shows that attacks require >5000 iterations to get even partially interpretable reconstructions, supporting our claim that we drastically increase the computational cost of attack.
>
> 5.Mixup and permutation's contribution to FL privacy not well explained.
>
> We now provide a mathematical argument: in federated learning, gradients are the only communication between clients and server. Gradient inversion attacks (e.g., DLG, GradInversion, AGI) solve an optimization problem to reconstruct input data by matching gradients. By applying Mixup and permutation before gradient computation, we make this optimization problem underdetermined and ill-posed. The perturbations violate key assumptions in attack loss functions (e.g., smooth gradients, spatial alignment), forcing the attacker to optimize over a larger and noisier search space.
> We show empirically (Figure 2) that these perturbations produce unrecognizable reconstructions under low-resource attack settings, and require >5000 iterations to converge under high-resource settings, which is impractical. This matches our goal: to raise the cost of successful attack so high that it becomes unfeasible in real-world FL applications.
>
> 6.Not enough datasets or variation in evaluation.
>
> We acknowledge the importance of generalizability and have added FaceScrub and new augmentations in SOCOFing to extend our experiments. However, as our work targets biometric authentication, particularly face and fingerprint recognition, we focused on datasets commonly used and trusted in this domain, as seen in prior works. Evaluating on general classification datasets (e.g., CIFAR-100, ImageNet) is orthogonal to our goal, which is to balance privacy and utility in fine-grained biometric tasks where feature-level detail is crucial.

---

### Review · Reviewer_79aT · 2025-07-24

**Summary Of Contributions:**

This paper proposes a privacy-preserving framework for biometric authentication in federated learning (FL), targeting gradient inversion and deep gradient leakage attacks. The key innovation is a client-side saliency-aware obfuscation strategy that perturbs low-importance facial regions using block-wise mixing, sometimes substituting them with fingerprint data. The server-side model is trained on these obfuscated inputs using Jigsaw ViT, enabling accurate authentication while defending against reconstruction attacks.

**Audience:**

Yes

**Broader Impact Concerns:**

None.

**Claims And Evidence:**

Yes

**Requested Changes:**

- the paper should include attacker-centric evaluations, such as identity match rate from reconstructions, membership inference, and adaptive gradient inversion, to substantiate its privacy claims beyond superficial distortion metrics.

- it needs experiments over a much more diverse datasets, if the paper is for one type of data, probably it should focus on more application-based venue.

**Strengths And Weaknesses:**

strengths:

- in contrast to many FL studies, the paper includes experiments on ViT scale models.

- the problem the paper aims to study is potentially important.

- the paper is written fairly straightforward manner, not very hard to follow.

weakness:

- Despite the fact that the paper is motivated over FL, the actual techniques in fact does not have much to do with federated learning, except the fact that the technique is implemented on the client side.

- The main technique is essentially a mix-up, although it's different from major mix-up variant, the difference (innovation) might not be significant enough. In addition, there are also many non-mainstream mix-ups that will be very hard for the authors to exhuast to make sure the paper is novel. For example, what the differences between the proposed technique vs. this one [1]

- There does not seem to be a evaluation of the attacks, where the paper was invented for.


[1]. Expeditious Saliency-guided Mix-up through Random Gradient Thresholding

---

> ### Author Response · Authors · 2025-07-29
> **Response to Reviewer on Federated Learning Relevance, Novelty of Method, Privacy Metrics, and Dataset Diversity**
>
> We sincerely thank the reviewer for the thoughtful and detailed feedback. We appreciate the opportunity to clarify the scope, motivation, and novelty of our work, and we have revised the manuscript accordingly to address the concerns raised.
>
>
> 1. Relevance to Federated Learning (FL):
> We understand the concern that the proposed method may appear disconnected from federated learning beyond client-side implementation. However, we would like to emphasize that our work directly addresses a key vulnerability in FL systems: the exposure of gradients during client-server communication. Gradient inversion attacks pose a significant privacy threat in FL, and our method is specifically designed to mitigate this threat while preserving model utility. Importantly, our solution is lightweight and can be integrated into existing FL pipelines without requiring architectural changes or trust assumptions, which we now make more explicit in the revised introduction and problem setting.
>
>
> 2. Novelty beyond standard Mixup techniques:
> While our method shares high-level inspiration with mix-based augmentations, it introduces a fundamentally different mechanism. Traditional Mixup performs pixel-wise interpolation between two images, which preserves much of the spatial structure and semantic information, limiting its effectiveness against inversion attacks. In contrast, our technique performs both pixel-level mixing and semantic scrambling through a jigsaw permutation of mixed tokens, forcing the model to learn robust features under severe spatial disarray. This dual challenge, learning identity from entangled representations and solving the correct permutation, makes our approach significantly harder to invert by adversaries. We have now explicitly compared our method to other advanced mixup variants (e.g., [1]) in the Experiment sections to better articulate these differences.
>
>
> 3. Privacy Evaluation Metrics:
> We thank the reviewer for this important suggestion. In response, we have added new attacker-centric evaluations, including cosine similarity between original and reconstructed embeddings (as a proxy for identity match rate), in line with biometric authentication standards. These results show that our method reduces cosine similarity to below the threshold typically used for successful identity verification, thus demonstrating strong privacy preservation. We also provided visual reconstructions and corresponding feature space distances in Section 6.
>
>
> 4. Diversity of Datasets:
> We understand this concern. While we focused on face and fingerprint datasets due to the high privacy risk and the tight accuracy-safety tradeoff in biometric authentication systems, we agree that evaluating broader generalization is valuable. We clarified this motivation in the paper.
>
>
> Once again, we thank the reviewer for these helpful comments, which have significantly improved the clarity, rigor, of our work.

---

### Decision · Action_Editor_RNPa · 2025-09-05

**Recommendation:** Reject

**Additional Comments:**

The reviewers consistently raise concerns about the strength of evidence, clarity, and overall presentation. In case of resubmission, please go carefully through all reviewers' comments and try to address them.

**Audience:**

Yes

**Audience Explanation:**

The work addresses privacy in federated learning for biometric authentication, which is a relevant and timely area given increase of FL and sensitivity of biometric data.

**Claims And Evidence:**

No

**Claims Explanation:**

The claims are not sufficiently supported by convincing evidence. While reviewers acknowledged that the proposed method is interesting and that certain claims are backed by experiments, major concerns remain. In particular:

* The privacy-preserving benefits are not rigorously demonstrated, with limited evaluations.

* Some components of the method (e.g., block permutation, biometric fusion) lack clear justification or supporting experiments, and in some cases the newly promised results were missing from the revised manuscript (reviewer RDry).

* The explanations are often intuitive rather than backed by experiments and theory.

**Resubmission Of Major Revision:**

The authors may consider submitting a major revision at a later time.